# The Activity of Isoquinoline Alkaloids and Extracts from *Chelidonium majus* against Pathogenic Bacteria and *Candida* sp.

**DOI:** 10.3390/toxins11070406

**Published:** 2019-07-12

**Authors:** Sylwia Zielińska, Magdalena Wójciak-Kosior, Magdalena Dziągwa-Becker, Michał Gleńsk, Ireneusz Sowa, Karol Fijałkowski, Danuta Rurańska-Smutnicka, Adam Matkowski, Adam Junka

**Affiliations:** 1Department of Pharmaceutical Biology, Wroclaw Medical University, Borowska 211, 50-556 Wroclaw, Poland; 2Department of Analytical Chemistry, Medical University of Lublin, Chodźki 4a, 20-093 Lublin, Poland; 3Departament of Weed Science and Tillage Systems, Institute of Soil Science and Plant Cultivation, Orzechowa 61, 50-540 Wrocław, Poland; 4Department of Pharmacognosy, Wroclaw Medical University, Borowska 211a, 50-556 Wroclaw, Poland; 5West Pomeranian University of Technology in Szczecin, Faculty of Biotechnology and Animal Husbandry, Department of Immunology, Microbiology and Physiological Chemistry, Piastów 45, 70-311 Szczecin, Poland; 6Pharmaceutical Microbiology and Parasitology, Wroclaw Medical University, Borowska 211a, 50-556 Wroclaw, Poland; 7Laboratory of Experimental Cultivation, Botanical Garden of Medicinal Plants, Wroclaw Medical University, Al. Jana Kochanowskiego 14, 50-556 Wroclaw, Poland

**Keywords:** isoquinoline alkaloids, antimicrobial activity, *Chelidonium majus*, Major isoquinoline alkaloids from *Chelidonium majus* are active against several pathogenic bacteria and yeast. Sanguinarine and chelerythrine are most efficient against Gram positive and Gram negative strains; respectively.

## Abstract

*Chelidonium majus* (*Papaveraceae*) extracts exhibit antimicrobial activity due to the complex alkaloid composition. The aim of the research was to evaluate the antimicrobial potential of extracts from wild plants and in vitro cultures, as well as seven major individual alkaloids. Plant material derived from different natural habitats and in vitro cultures was used for the phytochemical analysis and antimicrobial tests. The composition of alkaloids was analyzed using chromatographic techniques (HPLC with DAD detection). The results have shown that roots contained higher number and amounts of alkaloids in comparison to aerial parts. All tested plant extracts manifested antimicrobial activity, related to different chemical structures of the alkaloids. Root extract used at 31.25–62.5 mg/L strongly reduced bacterial biomass. From the seven individually tested alkaloids, chelerythrine was the most effective against *P. aeruginosa* (MIC at 1.9 mg/L), while sanguinarine against *S. aureus* (MIC at 1.9 mg/L). Strong antifungal activity was observed against *C. albicans* when chelerythrine, chelidonine, and aerial parts extract were used. The experiments with plant extracts, individually tested alkaloids, and variable combinations of the latter allowed for a deeper insight into the potential mechanisms affecting the activity of this group of compounds.

## 1. Introduction

Phytochemical characterization of the raw material enables one to recognize the production patterns of the plant compounds accumulated in response to the environmental factors. The composition and proportions of individual components are of great importance from the herbal drug usefulness point of view. Such studies should be conducted on a large number of samples due to the multitude of factors that may affect the phytochemical profile of plants. This can be done, *inter alia*, by comparing metabolite compositions in plants collected from natural habitats and in vitro cultures.

For our research, we have chosen a well-known medicinal plant *Chelidonium majus* L. (greater celandine). The species has a long tradition of being used mainly in folk medicine. It has been used in herbal medicine since Dioscorides and Pliny the Elder times, in the 1st century AD [1]. So far, anticancer, antimicrobial (antibacterial, antiviral, antifungal), antiprotozoal, anti-inflammatory, antispasmodic, spasmolytic, cholekinetic, muscle relaxant activities of *C. majus* extracts and several separated compounds have been reported [1]. The most abundant specialized metabolites produced in aerial and underground (roots and root collar, hereafter referred to as “roots”) parts of the plant are isoquinoline alkaloids, mostly derivatives of benzophenantridine (chelidonine, chelerythrine, sanguinarine) protoberberine (berberine, coptisine, stylopine), and protopine (protopine, allocryptopine). The bioactivity of individual alkaloids has been linked, among others, to the presence of a methoxy substitutions, the iminium bond, a charge due to a quaternary nitrogen atom, a methyl group bonded to the quaternary nitrogen atom or lack of this substitution [2,3,4,5,6,7,8]. The published results do not always coincide or complement each other, or they are contradictory. Despite many multidirectional studies, the complex alkaloid composition of the species is still unexplored, and so is the bioactive potential of the plant [9,10,11,12,13,14,15]. 

In this study, we evaluated antimicrobial activity of seven alkaloids and *C. majus* extracts from plants derived from natural habitats and in vitro cultures. A comparison of the alkaloid profile of extracts obtained from aerial parts and roots of plants collected from different habitats was also performed using chromatographic techniques. Moreover, antimicrobial activity of seven major alkaloids was tested and the results were correlated with alkaloid content.

## 2. Results

### 2.1. Comparison of Extraction Effectiveness

HPLC analysis showed the presence of seven major alkaloids of the three main classes (phenanthridine: chelidonine, chelerythrine, sanguinarine; protoberberine: berberine, coptisine; protopine: allocryptopine, protopine). Chemical structures of the compounds are presented in Figure 1. The content of these alkaloids in extracts obtained with the use of various solvents expressed as µg on g of dried plant material is shown in Figure 2, Figure 3 and Figure 4.

The aerial and underground parts differed significantly in the content of alkaloids (Figure 2 and Figure 3). The blooming herb after separating the fruits, was rich in chelidonine, while in the separated fruits coptisine was the most abundant alkaloid (Figure 5). Roots were generally a much alkaloid richer raw material. The content of all alkaloids was much higher in roots than in aerial parts. There was a particularly high content of sanguinarine, chelidonine, chelerythrine, and allocryptopine in the samples collected from all five habitats. The most abundant compound in roots was sanguinarine (1986.43 µg/g d.w.), and its highest content was recorded in the plants from habitat E (Szukalice, Lipowa). In aerial parts, coptisine was a predominant compound (857.29 and 4979.12 µg/g d.w). The highest content of coptisine was found in plants from habitat A (Wrocław, Kochanowskiego). 

The analysis across multiple test attempts using different concentrations of extraction solvent showed no tied ranks. The yield of all seven alkaloids was highest at the methanol concentration of 80%, compared to 70% and 90%. There were large differences between 70 and 80%, and much smaller between 80 and 90%. 

Methanol extraction of 80% was found to be the most effective in terms of alkaloids recovery (Figure 4). 

The representative chromatograms are presented in Figure 5.

Differences in the alkaloids content were also found among five different habitats (A–E). The content of individual alkaloids depends on the locality of the plant harvest, and it is different for individually analyzed alkaloids from roots and aerial parts evaluated separately.

### 2.2. MIC Evaluation

Extracts of aerial parts and roots of *C. majus* derived from natural habitats and in vitro cultures were used for the antimicrobial assays.

None of the applied extracts was active against *K. pneumoniae*, *P. aeruginosa* and *E. coli* strains, which are Gram-negative bacteria. 

In the case of Gram-positive pathogen *S. aureus*, MIC of roots MeOH extract was 62.5mg/L (Table 1). Also, a strong reduction of bacterial biomass (49%) in comparison to untreated bacteria was observed when 31.25 mg/L of this extract was used. 

Individually tested alkaloids displayed MIC against *S. aureus* and *P. aeruginosa* in the range between 1.9 and 125 mg/L, depending on the compound, and between 31.25–62.5 mg/L against *C. albicans*, depending on the compound (Figure 6). Chelerythrine was the most effective, of all tested compounds against *P. aeruginosa* (MIC at 1.9 mg/L), whereas sanguinarine was the most effective against *S. aureus* (MIC at 1.9 mg/L).

Berberine and allocryptopine were the least effective—reduction of 100% of *S. aureus* bacterial biomass was observed when 125 mg/L of compounds were tested. 

Strong fungal biomass reduction was observed for *C. albicans* when aerial parts extract was used, however no MIC was observed in analyzed range of concentrations. 

Only chelerythrine and chelidonine were active enough to allow calculation of MIC (31.25 and 62.5 mg/L, respectively) against *C. albicans*.

The MIC of combined sanguinarine-chelerythrine-chelidonine against *S. aureus* and *P. aeruginosa* was 3.12 and 6.25 mg/L, respectively, of mixture was used (Figure 7). Sanguinearine-chelerythrine and sanguinarine-chelerythrine-berberine mixtures were much more effective against *S. sureus* than against *P. aeruginosa* and *C. albicans* (Figure 7).

### 2.3. Cytotoxicity

Berberine, protopine and allocryptopine displayed no cytotoxic effect against L929 fibroblast cell lines in concentration of 15.6 mg/L; lack of cytotoxicity was observed for coptisine when 7.8 mg/L of this compound was introduced to L929 cell line; chelidonine and chelerythrinine displayed no cytotoxic effect in concentration of 3.9 mg/L. Cytotoxic properties were exhibited by sanguinarine which displayed adverse effects towards L929 cell line at concentrations of 500 µg/L or higher.

## 3. Discussion

The phytochemical profile that determines the biological activity of the raw material is difficult to characterize due to a large number of compounds produced in the plant and the multitude of factors determining their formation. For this reason, we compared the results of the bioactivity assays performed using plants harvested from different natural habitats, the in vitro cultures, as well as seven individual alkaloids and their mixtures. 

So far, both herb and *C. majus* root and root collars as underground parts were considered to be rich in alkaloids, and hence they were used, mainly in folk medicine [1]. Our research has shown that roots of the species form a much richer source of these compounds in terms of their quality and quantity. It has been noted for both the in vitro and in vivo plant material. In turn, a kind of variation within the resulting composition of each plant parts was observed among samples collected from different habitats (Figure 2 and Figure 3) as well as from different in vitro culture treatment [9]. 

MIC of methanol extracts of aerial parts and roots, as well as in vitro cultures of shoots and roots was observed against Gram-positive bacteria when 31.25–62.5 mg/L, and 125–250 mg/L were used, respectively. Sanguinarine was found to be the most abundant constituent in plant extracts (2–4 mg/g d.w. in intact plants and in vitro cultures. The amounts of other alkaloids were much lower in the extracts except coptisine (almost 5 mg/g d.w.). In individual tests, all seven alkaloids exhibited, to various extent, activity against Gram-positive strain. Sanguinarine was the most potent with MIC at 1.9 mg/L, followed by coptisine, protopine, chelerythrine, chelidonine, berberine, and allocryptopine. This kind of activity was previously reported in the earlier research on celandine alkaloids [18,19,20,21]. These experiments showed that chelerythrine and sanguinarine were more effective against these bacteria than chelidonine and berberine. In our study, sanguinarine exhibited several to several dozen times stronger activity than coptisine (MIC 31.25 mg/L), chelerythrine and chelidonine (MIC 62.5 mg/L), or allocryptopine and berberine (MIC 125 mg/L). Sanguinarine activity was also stronger than roots methanolic extract rich in the compound (1617.91–1986.43 µg/g d.w.). The extract was effective when the concentration of 62.5 mg/L was used. On the other hand, sanguinarine alone was slightly less active (MIC 1.9 mg/L) against Gram-positive bacteria than mixtures of sanguinarine with chelerythrine or sanguinarine with chelerythrine and berberine (MIC 1.56 mg/L). 

In Kokoška et al. (2002) [22] experiments with multidrug-resistant bacteria existing in surgical wounds and infections of critically ill patients, *C. majus* root ethanol extract was also found to be effective against Gram-positive bacteria (*S. aureus*, *Bacillus cereus*) (MIC 15.63 and 62.5 mg of dry plant material/ml, respectively), but was inactive against Gram-negative (*P. aeruginosa*). Also, the concentration of 62.5 mg of dry plant material/ml effectively inhibited *C. candida* in these experiments. The aerial parts of *C. majus* used in the study of Kokoška et al. (2002) [22] were inactive against any of the test microbes.

Other studies in which methanolic extracts of *C. majus* were used are also consistent with the results from our experiments. Methanol extracts from leaves and petioles of *C. majus* plants grown in nature, as well as in vitro cultures [23] were potent against Gram-positive, rather than Gram-negative strains. In these studies, methanolic extracts were examined against *Bacillus subtilis, Micrococcus luteus, Sarcinia lutea,* and *S. aureus*, *E. coli, Proteus mirabilis, Salmonella enteritidis*, and clinically isolated *C. albicans*. Both, in vivo and in vitro plant material extracts exhibited similar bioactivity. Only some extracts were comparable against *E. coli*, *S. enteritidis*, and *C. albicans* to reference antibacterial and antifungal drugs (streptomycin, bifonazole, respectively), whereas the rest of them showed low or no activity [23].

None of the extracts, as well as sanguinarine and berberine alone were active against a Gram-negative bacteria (*P. aeruginosa*). However, other individually tested alkaloids displayed various degrees of activity against *P. aeruginosa* with MIC ranging between 1.9 and 125 mg/L (Figure 6). Chelerythrine was found to be the most potent (1.9 mg/L), followed by allocryptopine, coptisine (62.5 mg/L), and protopine, chelidonine (125 mg/L). In other studies, chelerythrine was also able to eradicate Gram-negative bacteria (*P. aeruginosa*, *E. coli*, *Klebsiella pneumoniae*, *Salmonella gallinarum*, *S. typhi*, *S. paratyphi*, *Proteus vulgaris*, *Shigella flexneri*, *Vibrio cholerae*) [18,19,20,21,24,25]. Nevertheless, sanguinarine and berberine were also listed as effective against Gram negative strains in the mentioned studies, which was not corroborated by our results. 

The combined sanguinarine-chelerythrine-chelidonine were able to inhibit 100% of microbial inoculum when 6.25 mg/L of mixture was used. These results indicated that chelidonine may have an additional effect when combined with other alkaloids. It may be due to the chemical structure of chelidonine, which is different than sanguinarine and chelerythrine. Chelidonine is a benzoisoquinoline alkaloid with a tertiary nitrogen in the molecule, unlike sanguinarine and chelerythrine, which both contain a quaternary nitrogen atom (Figure 1) whose charge can depend on the pH also in the microenvironment of the bacterial cells. Various types of alkaloid bioactivity at the cellular level due to the differences in their chemical structure were well documented in the study of Barreto et al. (2003) [6]. In experiments with several groups of isoquinoline alkaloids on oxygen uptake in mouse liver mitochondria, these three alkaloids have shown a different scheme of action due to their chemical structure [6]. Generally, phenanthrene skeleton had a very low effect on oxygen uptake, while other building elements of individual alkaloids seemed to be important. Chelerythrine and sanguinarine, strongly inhibited succinate-dependent respiration and, to a lesser extent, malate–glutamate respiration, whereas chelidonine had no apparent effect. Allocryptopine, an uncharged molecule with a C=O group, similarly to chelidonine, was not effective. In Barreto et al. (2003) [6] study, the manner of compounds action was linked to the degree of substitution of a nitrogen atom in a molecule. In turn, berberine and coptisine, both with an unsubstituted quaternary nitrogen atom, have shown a marked inhibitory effect on malate-glutamate respiration and a smaller, although significant, effect on succinate respiration. The presence of a methyl group seems to be of less importance for the direction of biological activity. Chelerythrine, sanguinarine, and chelidonine contain methyl group but they present different activity pattern against microorganisms [26]. 

In case of antifungal assays, among seven individually tested alkaloids, only chelerythrine and chelidonine, as well as extracts from in vitro-derived shoots, were able to inhibit 100% of *C albicans* colony forming units (31.25, 62.5, and 500 mg/L, respectively). Five other alkaloids, their mixtures, as well as in vivo and in vitro plant extracts showed no MIC in the analyzed range of concentrations. However, a strong fungal biomass reduction was observed (Figure 6, Table 1). The observation of other researchers has shown similar results, only with chelerythrine, sanguinarine, and their derivatives being up to several times more effective against pathogenic fungi than chelidonine [26,27,28]. Extracts from roots and shoots cultured in vitro on two different standard media, namely MS and B5, exhibited varied effect against *C. albicans*. It was probably related to the differences in the composition of micro and macro elements, sucrose and vitamins between the two culture media.

Microbes, during their the long evolutionary journey developed a plethora of systems aiming to evade or neutralize antimicrobial agents. It concerns not only human or animal pathogens but also environmental strains which co-exist in complex, multi-species habitat in water or soil having contact with plants [29,30,31].

In turn, it was proven, that bacterial pathogens in nosocomial environment are able to actively pump biocides out from their cytoplasm using “efflux pump” systems. It is also known that these mechanisms may be of un-specific nature, i.e., efflux pump activation as result of presence of specific biocide may activate bacterial organism to pump out wide spectrum of other biocides. An increased resistance to chlorhexidine antiseptic and cross-resistance to colistin antibiotic following exposure to chlorhexidine in *Klebsiella pneumoniae* is one of the most studied examples of such a phenomenon [32]. Such a mechanism may explain results obtained by us and other researchers showing various levels of antimicrobial activity of alkaloids provided alone or in mixture.

## 4. Conclusions

The complex composition of *C. majus* alkaloids contained in plant extracts can manifest a wide spectrum of antimicrobial activity, arising from structural diversity of the compounds. Alkaloids from all parts of *C. majus* and from in vitro biomass may find an application in eradication of both Gram-positive and Gram-negative cocci. They are also promising against *Candida* pathogens. Further detailed studies are necessary to fully understand the mechanisms of activity associated with the chemical structures of isoquinoline alkaloids. The rational design of the composition of alkaloids should be envisaged to combat various microbes depending on the activity of individual components and for this to happen, the natural proportions of the alkaloids in plant matricies can be manipulated both by means of plant treatment and post-harvest processing of the crude biomass and extracts obtained thereof. In further studies on *C. majus*, the factors at work that orchestrate the alkaloid profile and their strain-specific activity should be elucidated.

## 5. Materials and Methods 

### 5.1. Plant Material

#### 5.1.1. Wild Growing Plants

The studied plants were collected from five different locations in Poland: A: (Wrocław, Kochanowskiego Street 51°07’01.6”N 17°04’26.7”E51.117121, 17.074088—28.05.2017), B: (Wrocław, Kosciuszki Street 51°06’08.0”N 17°02’08.9”E51.102220, 17.035811—30.05.2017), C: (Turawa 50°44’45.9”N 18°02’29.1”E50.746071, 18.041428—11.06.2017), D: (Wrocław, Borowska Street 51°04’42.0”N 17°01’51.9”E51.078325, 17.031080—25.05.2017), E: (Szukalice, Lipowa Street 51°00’07.4"N 17°00’40.4”E51.002062, 17.011213—22.05.2017). The habitats of plants were overshadowed roadsides, as well as forest edges and shrubbery. The maximum distance between the localities was up to 150 km and minimum was 3 km. 

#### 5.1.2. In Vitro Plant Material 

Plant material from in vitro cultures was used for the analysis and bioactivity assays. The in vitro cultures establishment, as well as extraction procedures for the phytochemical analysis, were presented previously [9]. 

### 5.2. Phytochemistry

#### 5.2.1. Reagents and Standards

Alkaloid standards such as protopine (purity ≥ 95), berberine (purity ≥ 95%), chelidonine (purity ≥ 95%), chelerythrine (purity ≥ 90%), and sanguinarine (purity ≥ 90%) were purchased from Extrasynthese (France) and allocryptopine, (purity ≥ 95), coptisine (purity ≥ 98%) from Sigma (St. Louis, MO, USA). Ammonium acetate, acetic acid, HPLC grade methanol (MeOH), and acetonitrile (ACN) were from Merck (Darmstadt, Germany). Water was deionized and purified by ULTRAPURE Millipore Direct-QVR 3UV-R (Merck, Darmstadt, Germany).

Alkaloid mixtures used for bioactivity assays were obtained from the following sources:

sanguinarine (Sang), chelerythrine (Cheler) were isolated as a mixture from *Coptis chinensis* rhizoma (19g/100g yield). Three mixures were used: Sang-Cheler (0.2:1 *w*/*w*); Sang-Cheler-Chelid (0.2:1:1, *w*/*w*); Sang-Cheler-Berb (0.2:1:1, *w*/*w*).

#### 5.2.2. Sample Preparation

Dried raw material was divided into aerial parts (stems with leaves, flowers and fruits) and underground (roots and root collars) parts and powdered with mortar and pestle. The extraction of intact plants was performed in round-bottom flasks with a solvent to solid ratio of 1:20 (*v*:*w*) in ultrasonic bath (3 × 15 min). Samples were extracted with methanol or ethanol and 50 mM hydrochloric acid according to the procedure by Kulp et al. [33]. Additionally, extraction was conducted with acidified (50 mM HCl) aqueous methanol in three different proportions (90:10, 80:20 and 70:30 *v*/*v*). The extracts were combined, evaporated and dissolved in 20 ml of methanol.

#### 5.2.3. HPLC Analysis

Chromatography was carried out using a VWR Hitachi Chromaster 600 chromatograph (Merck, Darmstadt, Germany) with a spectrophotometric detector (DAD) and EZChrom Elite software (Merck). The samples were analyzed on an XB-C18 reversed phase core-shell column (Kinetex, Phenomenex, Aschaffenburg, Germany) (25 cm × 4.6mm i.d., 5 μm particle size), kept at 25 °C. Mobile phase consisted of acetonitrile (A) and 10 mM water solution of ammonium acetate adjusted to pH 4 with acetic acid (B). Gradient elution program was as follows: from 0 to 20 min: 20% A; from 20.5 to 27 min 25% A and from 27.5–60 min. 30% A at the flow rate of 1 mL/min. Chromatograms were recorded in the range of wavelength from 220 to 400 nm. The identity of compounds in plant extracts was confirmed by comparison of retention times and spectra with corresponding standards. Peak homogeneity was established comparing the spectrum recorded at the three peak sections upslope, apex, and downslope with the reference spectrum. Additionally, the chromatographic fractions eluted at the retention time characteristic for the investigated alkaloids were collected using a Foxy R1 fraction collector (Teledyne Isco, Lincoln, NE, USA), and their identity was confirmed by direct injection mass spectrometry (micrOTOF-Q II, Bruker Daltonics, Bremen, Germany) using Compass DataAnalysis software version 4.1. The operating conditions were as follows: positive ionization mode, ion spray voltage: 4500 V; fragmentator voltage: −500 V, corona discharge: 4000 nA, flow rate of nitrogen: 3.5 L/min, temperature of nitrogen: 200 °C and evaporator temperature: 350 °C. The generated ions were analyzed in the range of 50–500 *m*/*z*.

Quantitative analyses were performed at following wavelengths: 290 nm for protopine, allocryptopine and chelidonine, 359 nm for coptisine, 329 nm for sanguinarine, 346 nm for berberine, 318 nm for chelerythrine. Validation of the method was performed in our previous study [10].

### 5.3. Statistical Analysis

To explore differences in treatments (the content of protopine, allocryptopine, chelidonine, coptisine, chelerythrine, sanguinarine, and berberine) across multiple test attempts (different concentrations of extraction solvent, different habitats of intact plants) the Friedman ANOVA analysis followed by the Dunn test and Bonferroni correction was performed. The statistical significance of differences between treatments was considered significant at *p* < 0.05. All statistical processing was performed using Microsoft Excel (Office 365, Microsoft, Redmont, WA, USA).

### 5.4. Experimental Design for Bioactivity Assays

To conduct the bioactive potential of *C. majus* extracts bioactivity assays against Gram-positive (*Staphylococcus aureus*), Gram-negative (*Pseudomonas aeruginosa, Klebsiella pneumonia, Escherichia coli*), and pathogenic yeast (*Candida albicans*).

#### 5.4.1. Strains

The following microbial strains from ATCC collection were used for experimental purposes: *Staphylococcus aureus* 6538; *Pseudomonas aeruginosa* 14452; *Klebsiella pneumoniae* 700603; *Escherichia coli* 25922. Clinical fungal strain: *Candida albicans* 10231.

#### 5.4.2. MIC Evaluation

Standard microdilution technique according to EUCAST guidelines was used to assess antimicrobial potential of the methanol extracts (from underground and aerial plant parts). Survival of cells subjected to extract’s activity was assessed by the TTC assay (based on ability of colorless triphenyl tetrazolium chloride (TTC) compound to change into red formazan in the presence of living microbes; qualitative technique) and using spectrophotometry (λ = 600; semi-quantitative technique).

## Figures and Tables

**Figure 1 toxins-11-00406-f001:**
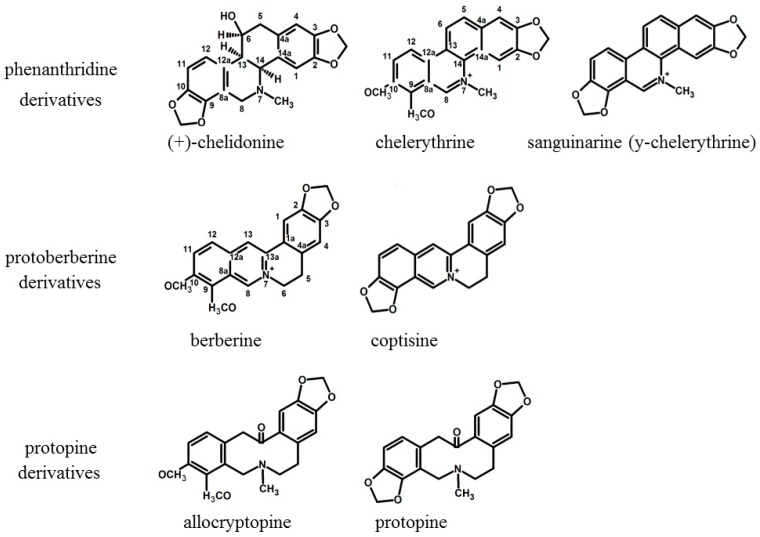
Structures of isoquinoline alkaloids present in *C. majus*.

**Figure 2 toxins-11-00406-f002:**
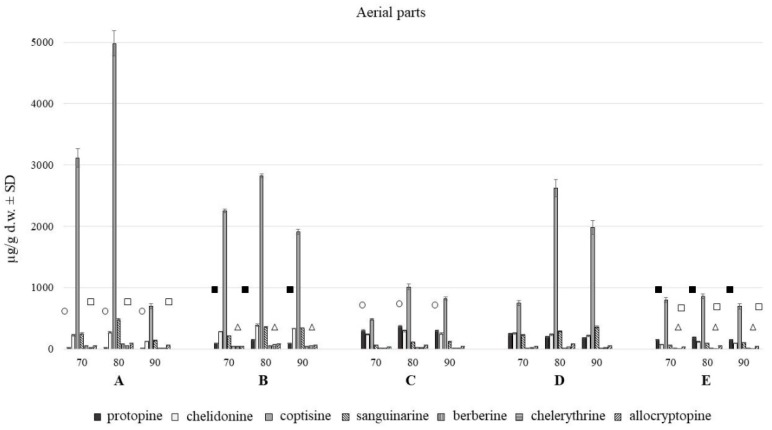
Isoquinoline alkaloids content in aerial parts with fruits of *C. majus* collected from different habitat (70, 80, 90% of methanol acidified with 50 mM HCl; **A**–**E**—different habitats). Statistically significant differences in the content of each compound between plants harvested from five different habitats are presented as marks of the same shape and color; protopine—white circle, chelidonine—black squares, chelerythrine—white triangles, allocryptopine—white squares, no marks—no statistically significant differences.

**Figure 3 toxins-11-00406-f003:**
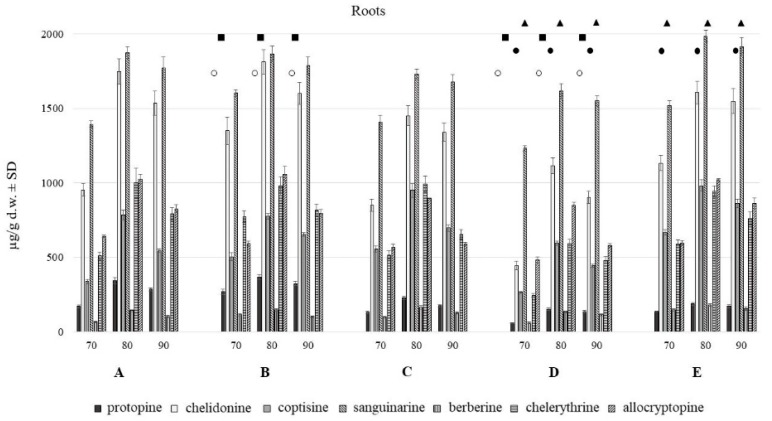
Isoquinoline alkaloids content in roots of *C. majus* collected from different habitat (70, 80, 90% of methanol acidified with 50 mM HCl; **A**–**E**—different habitats). Statistically significant differences in the content of each compound between plants harvested from five different habitats are presented as marks of the same shape and color; protopine—white circle, chelidonine—black squares, coptisine—black circles, sanguinarine—black triangles, chelerythrine—white triangles, no marks—no statistically significant differences.

**Figure 4 toxins-11-00406-f004:**
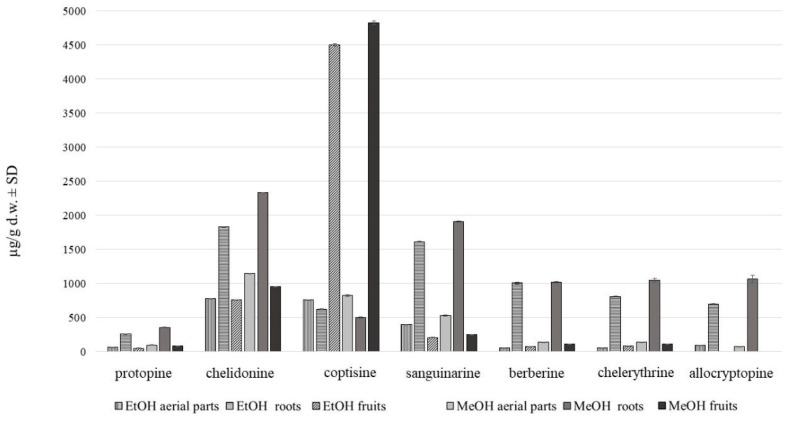
Alkaloid content in *C. majus* methanol and ethanol extracts of aerial parts and roots (µg/g d.w. ± SD) collected from place A.

**Figure 5 toxins-11-00406-f005:**
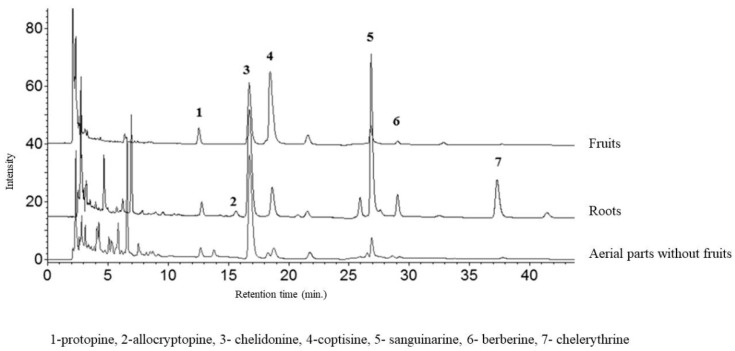
HPLC-DAD chromatogram of alkaloids (acquired at 290 nm) in extracts of *C. majus* fruits, aerial parts without fruits, and roots.

**Figure 6 toxins-11-00406-f006:**
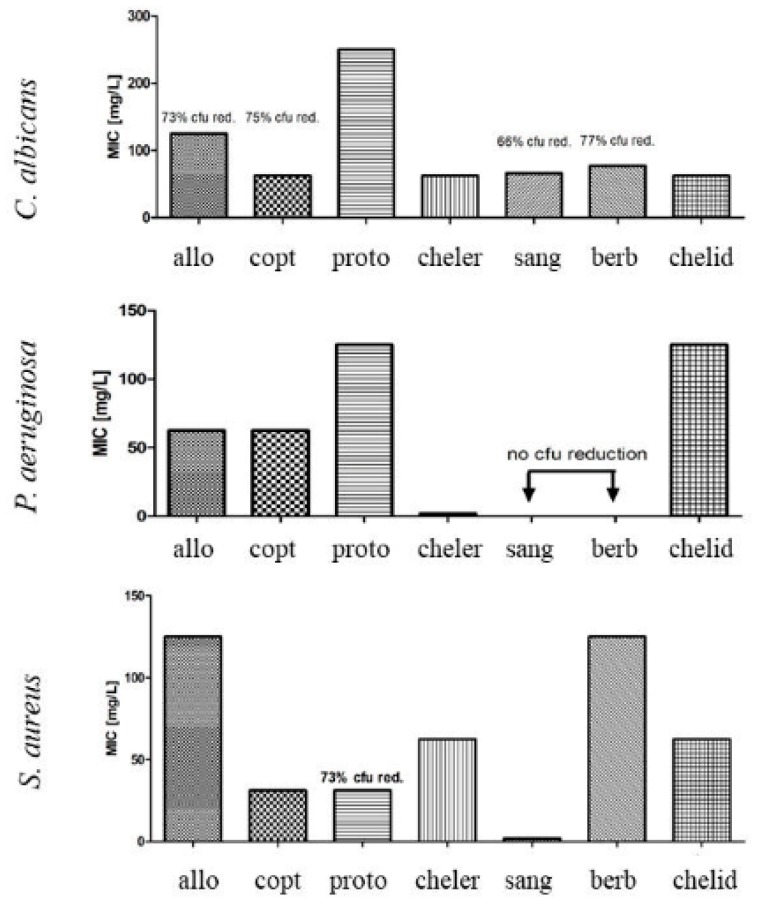
A comparison of Minimum Inhibitory Concentration (MIC) of the following alkaloids: allo = allocryptopine, copt = coptisine, proto = protoberberine, cheler = chelerythrine, sang = sanguinarine, berb = berberine, chelid = chelidonine.

**Figure 7 toxins-11-00406-f007:**
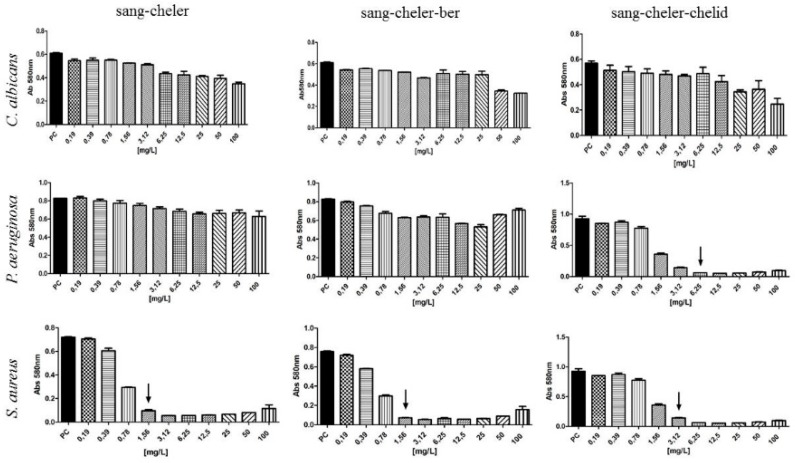
Antimicrobial activity of alkaloid mixtures. Arrows indicate MIC; sang = sanguinarine, cheler = chelerythrine, berb = berberine, chelid = chelidonine.

**Table 1 toxins-11-00406-t001:** Antimicrobial activity of methanolic extracts from intact plants and in vitro plant material. Asterisk indicates concentration which was able to inhibit 100% of microbial inoculum (i.e., Minimum Inhibitory Concentration—MIC (mg/L).); no asterisk next to value represents situation when at least 70% of microbial cell was reached; “minus” sign stands for very weak ability of extract to reduce colony-forming unit (cfu) number.

Solvent/Plant Material	*Candida albicans*	*Staphylococcus aureus*
MeOH aerial	−	−
MeOH roots	−	62.5 *
Culture medium/plant material	*Candida albicans*	*Staphylococcus aureus*
MS roots	125	500 *
MS shoots	500 *	500 *
MS+N roots	3.9	500 *
MS+N shoots	3.9	500 *
½ MS roots	−	−
½ MS shoots	−	−
1.5%suc. + N roots	−	−
1.5%suc.+ N shoots	500	250
1.5%suc. roots	250	125
1.5%suc. shoots	500	500 *
B5 roots	−	250 *
B5 shoots	−	500 *
B5 + N roots	−	−
B5 + N shoots	−	250 *

MS–standard culture medium [16]; ½ MS–medium with reduced macro-, and microelements concentration; B5–standard culture medium [17]; +N–supplementation with double amount of NH4^+^ ions and simultaneous depletion of an equivalent amount of nitrate ions; 1.5%suc–MS medium supplemented with 1.5% of sucrose.

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
