# Peer review of "The Activity of Isoquinoline Alkaloids and Extracts from Chelidonium majus against Pathogenic Bacteria and Candida sp."

_toxins, 2019, doi:10.3390/toxins11070406_

Round 1
Reviewer 1 Report
The paper entitled “The activity of isoquinoline alkaloids and extracts from Chelidonium maius against pathogenic bacteria and Candida sp.” describes a protocol for extraction and characterization of isoquinoloine alkaloids form C. maius and from cell culture. The extracts as well as the pure alkaloid standards have been then tested for their antimicrobial activity.
The paper shows several flaws:
1) The English used to draft the manuscript must be extensively reviewed by a native speaker to adjust lexicon and expressions. There are several mistakes spread across the text, which needs a serious editing.
2) The analytical method for extraction and characterization of isoquinoloine alkaloids form C. maius has been already described (Sowa I. et al., doi.org/10.1155/2018/9624327)
3) There are too many figures, that can be easily replaced by tables. In addition, these figures are not properly commented in the text. Figure 9, for example, has never been mentioned in result section.
4) Plants have been collected from different geographic areas. These areas are distant from 3 to 150 km. Is it significant? what about the statistical analysis of alkaloid content in plant from different geographic areas? which type of extract has been used for testing the antimicrobial activity? Table 1 is incomprehensible. The author should report the data in a different way.
5) Plant extract obtained by cultured cells shows different inhibitory effects against C. albicans according to the standard culture medium. Why? What is the difference between MS and B5 standard culture media?
6) Cytotoxicity. In Result section the authors state that “Berberine, protopine and allocryptopine displayed no cytotoxic effect against L929 fibroblast cell”. As a matter of fact, this is reported as a reference and must therefore be removed from Results.
Author Response
Responses to reviewers
Reviewer 1
The authors thank for all the valuable comments that have helped to improve significantly the manuscript.
Q 1: The English used to draft the manuscript must be extensively reviewed by a native speaker to adjust lexicon and expressions. There are several mistakes spread across the text, which needs a serious editing.
Answer: The manuscript has been extensively reviewed and improved by native speaker, who deals with professional reviewing and translating documents.
Q 2: The analytical method for extraction and characterization of isoquinoloine alkaloids form C. maius has been already described (Sowa I. et al., doi.org/10.1155/2018/9624327)
Answer: Yes, the analytical method has been described in our previous study (Sowa et al. 2018), that was why it has been mentioned as a reference in the Material and Methods section. Part of this analytical method together with the new necessary data on the plant material and solvent concentrations has been presented in the current manuscript, so that all information was clearly presented to the reader.
Q 3: There are too many figures, that can be easily replaced by tables. In addition, these figures are not properly commented in the text. Figure 9, for example, has never been mentioned in result section.
Answer: Thank you for this valuable comment. The number of figures has been reduced to the necessary minimum. All figures have been first cited and then presented in the manuscript in the proper order.
Q 4: Plants have been collected from different geographic areas. These areas are distant from 3 to 150 km. Is it significant? what about the statistical analysis of alkaloid content in plant from different geographic areas? which type of extract has been used for testing the antimicrobial activity? Table 1 is incomprehensible. The author should report the data in a different way.
Answer: Thank you for this comment. Table 1 has been deleted from the manuscript. The differences were presented graphically on the Figure 2 and 3. Statistically significant differences in the content of each compound between plants harvested from five different habitats have been presented as marks of the same shape and color. Additionally, an explanatory description has been placed under the figures.
Q 5: Plant extract obtained by cultured cells shows different inhibitory effects against C. albicans according to the standard culture medium. Why? What is the difference between MS and B5 standard culture media?
Answer: Thank you for this valuable comment. Undoubtedly, the composition of culture media affected the morphogenetic response of shoots and roots cultured in vitro, as well as the content of alkaloids produced in this plant material. These observations were presented in our previous study on the content of isoquinoline alkaloids in in vitro cultures of Chelidoniu majus (Zielinska et al. 2018). Standard culture media, namely MS and B5 differ in the composition of micro and macro elements, sucrose and vitamins. Extracts from roots and shoots in vitro exhibited varied effect against C. albicans probably due to the differences in the phytochemical profile of plant material cultured on media containing different concentrations of micro and macro elements, as well as other ingredients. Additionally, a suitable comment was placed in the discussion section.
Q 6: Cytotoxicity. In Result section the authors state that “Berberine, protopine and allocryptopine displayed no cytotoxic effect against L929 fibroblast cell”. As a matter of fact, this is reported as a reference and must therefore be removed from Results.
Answer: The reference data have been deleted from the Cytotoxicity subsection of the Results section.
Reviewer 2 Report
This article deals with the evaluation of the antimicrobial activity exhibited by extracts from the Chelidonium majus L. medicinal plant, both as wild material and as in vitro cultures. Seven major individual alkaloids (of the isoquinoline type), typically produced by the Chelidonium majus species, were also investigated in the screening and the composition of the various extracts was analyzed by reversed-phase HPLC-DAD under gradient elution.
The study is well presented, and the results obtained do deserve publication in Toxins. However, I would like to report the following minor items to be revised by the authors:
(1) Page 1, title: the plant name is Chelidonium majus; please replace maius by majus
(2) Page 1, abstract, line 6: the family name of the plant (namely, Papaveraceae) should be given close to the plant name
(3) Page 1, abstract, line 8: please add the word “individual” between major and alkaloids
(4) Page 1, abstract, line 10: after “chromatographic techniques” at the end of the line type, in brackets, the following text: (HPLC with DAD detection)
(5) Page 1, abstract, line 11: please delete the comma after the word “shown”
(6) Page 1, abstract, lines 17-18: please replace “their mixtures” by “variable combinations of the latters”
(7) Page 2, Figure 1: please check the word “derivatives” for the phenanthridine class
(8) Page 5, Table 2: I strongly suggest removing the last three columns in the table because they are meaningless with all those “minus” signs: the authors have already claimed in the text that “any of applied extracts was active against K. pneumoniae, P. aeruginosa and E. coli strains that are Gram-negative bacteria” (lines 96-97 of the manuscript)
(9) Page 5, lines 101-105: in the discussion of the MIC results, Figure 6 should be cited before Figures 7 and 8
(10) Page 8: Figure 9 must be first cited and after presented to the reader
(11) Page 9: Figure 10 must be first cited and after presented to the reader; furthermore, in the caption it would be better reporting the following legend: sang = sanguinarine; cheler = chelerythrine, and so on. Please avoid the use of capital letters for natural products
(12) Page 9, line 142: it is not clear what the authors mean for “concentrations than 500 ug/L or higher”
(13) Page 11, line 216: the authors have cited reference 26, but ref. 25 was never mentioned in the all text
(14) Page 12, reagents and standards, the purity of the isoquinoline alkaloids used in the study must be reported close to each standard
(15) Page 12, line 268: check the word “puri6ed”
(16) Page 12, line 279: the study by Kulp is included in reference 34
(17) Page 12, lines 291 and 295: check the word “con6rmed”
(18) Page 12, lines 291 and 296: the conditions for mass spectrometry are missing. I strongly suggest adding such missing data
Author Response
Responses to reviewers
Reviewer 2
The authors thank for all the valuable comments that have helped to improve significantly the manuscript.
All the grammar, stylistic and typo mistakes have been corrected and the whole manuscript has been reviewed and improved by native speaker, who deals with professional reviewing and translating documents.
For the standard deviation calculations, the number of repetitions was three.
For the antimicrobial activity assays of methanolic extracts the number of repetitions was three as well.
The list of references has been improved as requested.
The number of figures has been reduced to the necessary minimum, as it was required by one of the reviewers. All figures have been first cited and then presented in the manuscript in the proper order.
Reviewer 3 Report
Study of antimicrobial activity of alcaloids and Chelidonium maius extracts is actual. The article, material and methodology, as well as the results and discussion are very interesting and important not only for the researchers and new results, but also from practical point of view. After revision, mentioned below, I recommend to publish article.
Errors and comments:
Line 3 In title of article change the “maius” to majus
Line 63 “chelidonine.” remove dot
Line 69 “C. majus” C. majus
Fiugure 2 Figure 3 and Figure 4 – add from how many time did you repeated treatment to present the effect with standard deviation.
Line 86 “and roots at 290 nm. .” and roots at 290 nm.
Line 96 “K.pneumoniae” correct K. pneumoniae
Line 96 “E.coli” correct E. coli
Line 98 “S.aureus” correct S. aureus
Line 108 “C.albicans” correct C. albicans
Line 112 “in vitro” correct in vitro
Table 2 – number of repeated testing
Line 124 “MIC. .” correct MIC.
Line 136 – 137 “in vitro” correct in vitro
Line 160 “strains.” Only one gram-positive strain was used correct
Line 259, 385 “In vitro” correct In vitro
Line 310 Enterococcus faecalis – results absent in article but mentioned in material and methods!
Lines 328, 349, 357, 360, 372, 384 “Chelidonium majus” correct Chelidonium majus
Lines 333-334 Citation 3. is not complete
Line 381 “Kokoska” correct Kokoška
Lines 389 – 390 Make clear references and correct them in the text!
26. Kedzia, B.; Hołderna-Kedzia, E. The effect of alkaloids and other
27. groups of plant compounds on bacteria and fungi. Post. Fitoter. 2013, 1, 8–16
Author Response
Responses to reviewers
Reviewer 3
The authors thank for all the valuable comments that have helped to improve significantly the manuscript.
All the grammar, stylistic and typo mistakes have been corrected and the whole manuscript has been reviewed and improved by native speaker, who deals with professional reviewing and translating documents.
Thank you for indicating all the missing necessary expressions that we have introduced in the abstract and manuscript. We, hopefully improved the manuscript as requested.
The list of references has been improved as requested.
The number of figures has been reduced to the necessary minimum. All figures have been first cited and then presented in the manuscript in the proper order as requested.
All names of the natural products have been rewritten with lower case letter.
The conditions for mass spectrometry have been presented in the Materials and Methods section as requested.
Round 2
Reviewer 1 Report
The manuscript has been improved according to the suggestions.
I recommend to accept it.